# Monitoring the Circulation of SARS-CoV-2 Variants by Genomic Analysis of Wastewater in Marseille, South-East France

**DOI:** 10.3390/pathogens10081042

**Published:** 2021-08-17

**Authors:** Nathalie Wurtz, Océane Revol, Priscilla Jardot, Audrey Giraud-Gatineau, Linda Houhamdi, Christophe Soumagnac, Alexandre Annessi, Alexandre Lacoste, Philippe Colson, Sarah Aherfi, Bernard La Scola

**Affiliations:** 1Microbes, Evolution, Phylogeny and Infection (MEPHI), UM63, Institut de Recherche pour le Développement (IRD), Aix-Marseille University, 13005 Marseille, France; nathalie.wurtz@univ-amu.fr (N.W.); oceane.revol@etu.univ-amu.fr (O.R.); Audrey.GIRAUD-GATINEAU@ap-hm.fr (A.G.-G.); PHILIPPE.COLSON@UNIV-AMU.FR (P.C.); 2IHU Méditerranée Infection, 13005 Marseille, France; 3Assistance Publique-Hôpitaux de Marseille (AP-HM), Aix-Marseille University, 13005 Marseille, France; priscilla.jardot@ap-hm.fr (P.J.); Linda.Houhamdi@ap-hm.fr (L.H.); 4Bataillon de Marins-Pompiers de Marseille, 13003 Marseille, France; christophe.soumagnac@bmpm.gouv.fr (C.S.); alexandre.annessi@bmpm.gouv.fr (A.A.); alexandre.lacoste@bmpm.gouv.fr (A.L.)

**Keywords:** COVID-19, SARS-CoV-2, next-generation sequencing, variants, wastewater, sewage

## Abstract

The monitoring of SARS-CoV-2 RNA in sewage has been proposed as a simple and unbiased means of assessing epidemic evolution and the efficiency of the COVID-19 control measures. The past year has been marked by the emergence of variants that have led to a succession of epidemic waves. It thus appears that monitoring the presence of SARS-CoV-2 in wastewater alone is insufficient, and it may be important in the future to also monitor the evolution of these variants. We used a real-time RT-PCR screening test for variants in the wastewater of our city to assess the effectiveness of direct SARS-CoV-2 sequencing from the same wastewater. We compared the genome sequencing results obtained over the large RS network and the smaller B7 network with the different distributions of the variants observed by RT-PCR screening. The prevalence of the “UK variant” in the RS and B7 networks was estimated to be 70% and 8% using RT-PCR screening compared to 95% and 64% using genome sequencing, respectively. The latter values were close to the epidemiology observed in patients of the corresponding area, which were 91% and 58%, respectively. Genome sequencing in sewage identified SARS-CoV-2 of lineage B.1.525 in B7 at 27% (37% in patients), whereas it was completely missed by RT-PCR. We thus determined that direct sequencing makes it possible to observe, in wastewater, a distribution of the variants comparable to that revealed by genomic monitoring in patients and that this method is more accurate than RT-PCR. It also shows that, rather than a single large sample, it would be preferable to analyse several targeted samples if we want to more appropriately assess the geographical distribution of the different variants. In conclusion, this work supports the wider surveillance of SARS-CoV-2 variants in wastewater by genome sequencing and targeting small areas on the condition of having a sequencing capacity and, when this is not the case, to developing more precise screening tests based on the multiplexed detection of the mutations of interest.

## 1. Introduction

In the context of the COVID-19 pandemic caused by the Severe Acute Respiratory Syndrome CoronaVirus 2 (SARS-CoV-2), which appeared at the end of 2019 in China, global health situation is of great concern [1]. In July 2021, more than a year after the start of the pandemic, the disease had affected more than 190 million people and was responsible for more than 4.1 million deaths [2]. The main symptoms of COVID-19 are respiratory symptoms such as a cough or runny nose that can, in severe cases, result in acute respiratory distress syndrome, as well as gastrointestinal symptoms such as diarrhoea and vomiting, in 2–10% of cases [3,4]. Independently of the gastrointestinal symptoms, several recent studies have reported the presence of SARS-CoV-2 RNA in stool samples and in anal/rectal swabs, not only in symptomatic but also in asymptomatic patients and, consequently, in wastewater [5,6,7,8]. It has even been shown that viruses present in stool samples but not, to date, in wastewater are still infectious [9,10].

The detection of the virus in sewage offers the possibility of using wastewater as an epidemiological tool to monitor the invasion, prevalence, molecular epidemiology and potential eradication of the virus within a community. The detection and monitoring of SARS-CoV-2 in wastewater samples has already been reported in many countries [11,12,13,14,15,16,17,18,19,20,21]. The advantages of this method of epidemic surveillance are the possibility to monitor particular establishments, such as nursing homes or specific districts of a city [22], and even to assess the effects on health of the lockdown measures implemented in the context of the COVID-19 pandemic independently of the population testing [23]. A complex correlation between the copy number of SARS-CoV-2 RNA in wastewater and the number of newly infected persons has been identified [23].

The emergence of new variants of SARS-CoV-2, which have some mutations associated with increased transmissibility, a lower antibody response or both combined, has recently gained attention [24,25,26]. In the last months, several variants have spread around the world, particularly those known as the United Kingdom, Brazilian and South-African variants (hereinafter referred to as UK, Br and SA) and, more recently, new Indian variants (see below for the Nexstrain, WHO and Pangolin classifications of all the variants), which are the most representative and appear to spread with the greatest efficiency [27,28,29,30]. Besides the overall monitoring of the number of infected individuals, monitoring the relative frequency of these variants within a community is necessary to understand the dynamics of their transmission and to take appropriate public health actions, especially when the variants begin to spread. This surveillance is currently mainly conducted using whole-genome sequencing from nasopharyngeal samples of infected patients. This approach, although efficient, is expensive and labour- and time-consuming. It also presents major biases, since only genomes of symptomatic patients who have gone to hospital or to a laboratory to perform a RT-PCR screening test are analysed. A recent study suggests that at least one-third of SARS-CoV-2 infections are asymptomatic, which adds yet another bias [31]. The fact that infections by one or more of these variants are less symptomatic than others is enough to have a biased vision of their real circulation. Detecting variants via wastewater could, therefore, be a more efficient, faster and a more representative indicator for monitoring the emergence and spread of variants within a community.

Previous studies have shown that the shotgun sequencing of wastewater can provide information on several viruses simultaneously [32,33,34] and enables resolved genomic [35] and phylogenetic analyses [36,37]. Only a few recent studies have identified known genotypes of SARS-CoV-2, in addition to as-yet-unobserved variants from RNA recovered from the sewers [38,39,40,41,42,43], but these studies remain very marginal. Performing the detection of the variants using genome sequencing in multiple sewage water samples is also probably unaffordable in many regions, which are poorly equipped with a sequencing capacity. Several companies, as presented above, are beginning to offer RT-PCR multiplex systems that are expected to screen for the presence of the majority of the SARS-CoV-2 variants. In this context, and in order to evaluate whether this kind of system could help with monitoring the variants, SARS-CoV-2 sequencing from wastewater in the city of Marseille was initiated. The analysis by a RT-PCR multiplex screening system of the wastewater collected in a city district that had a different pattern than the city as a whole allowed us to compare it to a genomic sequencing method.

## 2. Results

### 2.1. Quantification of Variants in Wastewater Using Bio-T Kit^®^ SARS-CoV-2 UK and N501Y Variants (Biosellal, BIOTK125, Dardily, France)

The Bio-T Kit^®^ SARS-CoV-2 UK and N501Y variants (Biosellal, BIOTK125, Dardily, France) were evaluated on 31 SARS-CoV-2 RNAs corresponding to known genotypes. The results are presented in Appendix A and show that all genotypes of SARS-CoV-2 were correctly identified by the Bio-T Kit^®^.

In the large separate sewer (RS) and its subset (B7), wastewater networks, the distribution of the SARS-CoV-2 variants analysed using the Bio-T Kit^®^ showed that 70% of the SARS-CoV-2 detected in the RS network should correspond to the UK variant and 30% to other SARS-CoV-2 variants that did not have the mutation N501Y. In the B7 network, 8% of the SARS-CoV-2 detected should correspond to the UK variant; 35% to variants possessing the N501Y mutation except the UK variant (for example, SA, Br or Marseille 501/A.27) and 57% to other SARS-CoV-2 variants.

### 2.2. Variant Analysis by Sequencing of SARS-CoV-2 Present in Sewage Samples

Sequencing of the first sample corresponding to the RS network provided 1,435,398 and 2,317,230 reads for pools 1 and 2, respectively. Only 82% of the total reference genome length was covered when the sequencing reads from both pools 1 and 2 were mapped onto it, with a mean depth of 5582. The presence and proportion of each variant was determined in two steps. First, we evaluated the percentage of each mutation of interest found in the reads based on the mutation patterns of the variants (Table 1 and Appendix A). Then, we evaluated the number of detected representative mutations compared to the total number of possible mutations for each variant. In the RS sample, we identified mutations corresponding to four possible variants with read percentages ranging from 92% to 98%. The mutation patterns detected included mutation A23063T, corresponding to the N501Y substitution in the spike protein and present in UK variant but, also, in the SA and Br variants, as well as in a variant previously detected in our laboratory and referred to as Marseille-501 (Pangolin lineage A.27; Nextstrain clade 19B) [44]. For three variants, only this mutation was present, while, for the UK variant, we detected 7/21 hallmark mutations. Significantly, the region of the amino acid deletion del69-70 in the spike protein (nucleotide positions 21,767–21,772) was not covered by the sequencing reads (no read could be mapped on this region) and, thus, could not be interpreted. Accordingly, we can conclude that, based on the analysis of the unique sequences in the viral genome, the UK variant represented a mean of 95% of the sequences generated from the RS sewage sample and that no other circulating variant was clearly identifiable.

The sequencing of the second sample corresponding to the B7 network provided 1,422,464 and 2,573,568 reads for pools 1 and 2, respectively. A total of 91% of the total number of reads were mapped against the SARS-CoV2 reference genome. The average depth was 6086, and 84% of the total length of the reference genome was covered. A total of 13 mutation patterns were detected in this sample. The mutation A23063T (substitution N501Y) present in the UK, SA, Br and Marseille-501 variants was detected in 79% of the reads. Six other hallmark mutations of the UK variant (C3267T, C5388A, C23271A, T24506G, G24914C and A28111G) were detected at frequencies varying from 46% to 94%. Notably, four other hallmark mutations (G1006T, C6285T, G23593C and T24224C) of a variant named Marseille-484K.V3 (Pangolin lineage B.1.525) were detected at frequencies varying from 12% to 50% (Figure 1 and Appendix A). Two additional nonspecific mutations, G23012A (corresponding to substitution E484K in the spike protein) harboured by the Br, SA, B.1.525 and B.1.1.318 variants and T26767C harboured by the Marseille-484K.V4, B.1.525, B.1.1.318 and Indian (B.1.617.2) variants, were also detected in 20% and 12% of the reads, respectively. The signature mutations of eight different variants were identified. However, more than two signature mutations were identified for only two variants, which were the UK variant with 7/21 mutations ranging in frequency from 46% to 94% (mean = 64% per unique mutation) and the B.1.525 variant with 5/23 mutations ranging in frequency from 12% to 50% (mean = 27% per unique mutation).

### 2.3. Comparison of Variant Analyses in Wastewater and in People Newly Infected with SARS-CoV-2- in Marseille

The proportions of variants detected between 22 April and 2 May 2021 in people living in Marseille (corresponding to the global RS network) and those in people living in the B7 network perimeter are presented in Table 1. The proportion of UK variants observed in the population (90%) was closer to that of sewage genome sequencing (95%) than that obtained using the Bio-T Kit125^®^ screening test (70%).

In people living in the B7 area, the SARS-CoV-2 UK and B.1.525 variants represented 58% and 37% of the cases, respectively. The proportion of patients with UK variants observed in B7 was more in line with that of sewage genome sequencing (64%) than that obtained using the Bio-T Kit125^®^ screening test (8%). In addition, the proportion of the B.1.525 variant in the population of B7 (37%) aligned well with the mean value of 27% obtained by sewage genome sequencing. The Bio-T Kit125^®^ screening test classified this variant among the “other” variants; thus, its accuracy was difficult to evaluate in this case. However, on the basis of the evidence, the presence of 35% non-UK—N501Y variants (Br and SA variants) is considered to be true in spite of whether the results of the tests on the selected strains were correct (Appendix A).

## 3. Discussion

Recent studies have begun to identify known genotypes of SARS-CoV-2, as well as the variants recovered from sewers, but these studies remain scarce [38,39,40,41,42,43]. In this context, genotyping by sequencing SARS-CoV-2 from wastewater in the city of Marseille was undertaken using different methods, and the results correlated with the results of sequencing in patients.

During their routine survey of the distribution of SARS-CoV-2 variants in wastewater in different city districts using the Bio-T Kit^®^ SARS-CoV-2, BMPM identified that one district of the city (district B7, which drains part of the 12th and 4th districts of the city of Marseille) had a different profile compared to that of the general Marseille sewage network (RS). This assay was based on the unique detection of mutation N501Y and deletion Δ69-70 in gene S to separate SARS-CoV-2 into the UK variant, the Br/SA variants and others. Using this screening method, in the general RS network, 70% of the SARS-CoV-2 variants detected corresponded to the UK variant and 30% to variants that did not possess the mutation N501Y. In the B7 district, only 8% corresponded to the UK variant, 35% to other variants with mutation N501Y (except the UK variant) and 57% to other variants.

This clear difference in detection allowed us to test a sequencing method on wastewater. We searched for known variants using a matrix of the full mutation patterns specific to each SARS-CoV-2 variant. For each sample, we searched for the presence of mutations and, then, for the percentage of reads that harboured them. Ultimately, the number of signature mutations retrieved was small. In RS, we were able to identify seven signature mutations. The mutation A23063T (N501Y) was the only mutation that putatively corresponded to four different variants. The other mutations, present at frequencies ranging from 92% to 100%, were specific to the UK variant and were present in a mean of 95% of the sequences. This was, broadly speaking, in line with the Bio-T Kit^®^ SARS-CoV-2 screening (70% of the UK) but closer to the concomitant variants circulating in patients (91% of the UK). In B7, the picture was different and more complex, as there were 13 mutations detected corresponding to eight putative variants. However, more than two mutations were detected only for the UK and B.1.525 variants. By considering the mutations that are unique to these two variants, their mean frequencies were 64% and 27%, respectively. This is totally divergent from the results of the Bio-T Kit^®^ SARS-CoV-2 screening, which retrieved only 8% of UK, 34% of SA/Br and 57% of other variants. These results were more closely aligned with the actual circulation observed in patients for this district (58% of UK and 37% of B.1.525). The lower quality of discrimination can be due to the fact that we had a too-small panel of positive patients in B7 as compared to the huge number of typed patients in RS. It seems clear that, when a variant is largely predominant like the UK in the RS network, it tends to mask the other genotypes present. We have no certainty about the reason for this particular epidemiology in B7, only hypotheses. The main one is that it is a cluster linked to individuals who frequent the same place (market, restaurant, place of worship, sports hall, etc.). For example, after the end-of-the-year celebrations, we had a caricatural cluster of beta variants in an area of the city that we had easily linked to a group return of many holidaymakers returning from the Comoros. However, in the case of the B7 area, the anonymisation of the data did not allow us to contact all the individuals to trace the contamination. However, in the future, this is what should be done by first submitting a request to the ethics committee in this regard.

We were thus able to observe a comparable distribution among patients from the whole city of Marseille and in the RS wastewater network and a comparable distribution among patients living in the district corresponding to B7 and in the B7 wastewater network. More general, very recently published national studies carried out in the Netherlands and Belgium and several nations/cities in the UK have reached the same conclusions [42,45,46]. The results using the Bio-T Kit^®^ SARS-CoV-2 screening kit were less precise. It seems obvious that targeting one or two mutations by PCR cannot precisely detect the variants if they are not widespread. The PCR approach remains highly effective on well-targeted variants, as we have determined in our local setting by successively implementing real-time RT-PCR specific to the N501Y-harbouring variants and of the Marseille-4 and Marseille-1 variants [47,48,49]. It is likely that, in the future, molecular screening based on PCR will have to benefit from a multiplexing strategy capable of targeting several mutations simultaneously and will have to continuously adapt to the constant increasing viral diversity and complexity of mutation patterns of circulating variants. Additionally, PCR screening was performed here on a mix of different several strains and not on only one viral strain, as for the clinical samples. Multiplex RT-PCR could target different regions of multiple strains. However, by using the NGS approach, the depth of sequencing provided by the NOVASEQ technology seems able to increase the sensitivity of detection of some minority variants present in the samples, resulting in a more representative picture of the variants circulating in populations. The proportion of those obtained for each mutation pattern of all the variants is a more relevant and comprehensive picture of the complexity of the mix of the strains present in one sewage sample.

The use of wastewater as a tool to monitor the epidemiology and diversity of SARS-CoV-2 offers many advantages. Sewage samples are easy to collect, the sampling bias toward sequencing only symptomatic patients who have attended hospital or a laboratory to perform a RT-PCR screening test does not occur and fewer samples are required to determine the changes in viral infections [50]. To date, several variants have spread around the world, particularly the UK, Br and other variants and, most recently, the Indian variant, which have been the most prevalent and which appear to spread with a greater efficiency. Monitoring the spread of these variants in a city, community or, more precisely, in a district through wastewater will, therefore, improve the understanding of the dynamics of virus transmission, as well as help public health authorities to take more targeted measures rather than regional or national actions.

In this paper, we demonstrate that the analysis of SARS-CoV-2 variants from wastewater by next-generation sequencing is a powerful tool capable of improving our understanding of the outbreak transmission dynamics and, also, of reducing the burden of COVID-19. Rios et al. recently showed efficient SARS-CoV2 variant monitoring in the wastewater network of Nice, France by using Nanopore technology [45]. Herein, we demonstrated the key importance of the optimisation of storage and the pre-treatment conditions combined with a sequencing technology (NOVASEQ) providing a great depth of sequencing and, thus, being able to detect minority variants. The results obtained by these combined approaches properly correlate with the circulating strains in populations, as shown here with 1197 patients. A factor that may complicate the analysis of the genome data in sewers is the particularly high viral load of some variants that may cause some to be overstated. For example, patients infected with the delta variant have viral loads much higher than the previous variants [51], and in fact, since mid-July, we have observed an unprecedented level of SARS-CoV-2 copy numbers in wastewater, while the number of cases, even if it is high, has not reached our highest previous values (unpublished data). This means that the determination of the genome in the sewers will probably be more efficient in spotting clusters than in determining the real proportion of the circulating variants, unless we manage to implement a very complex standardisation system as soon as we have more than two or three variants in circulation.

In conclusion, we have demonstrated the value of monitoring the circulation of the variants by genomic sequencing in wastewater. This work suggests, in particular, that, rather than carrying out mass analyses of the network, such as RS, it will be preferable to test this network at many points, as we did for B7, to get an idea of the local circulation of the different variants and to better identify the clusters. This should help to identify the emergence of variant clusters as closely as possible in order to target the control measures and avoid catastrophic mass restrictions, which can have devastating psychological and economic effects. It should even be possible to test the reserves of black water from boats or planes to conduct mass screening of the variants while using minimal sampling.

## 4. Materials and Methods

### 4.1. Wastewater Collection

The study began on 22 April 2021 in the city of Marseille in the southeast of France. The SERAMM (Marseille Metropole Sanitation Department) sampled 250 mL of wastewater from the separate sewer networks (herein referred to as “RS”) by using an automatic sampler, the “ASP-Station 2000 RPS20B” (Endress Hauser, Huningue, France). This RS network drains the majority of Marseille’s wastewater and nearly all hospitals in the city, notably those dedicated to COVID-19 treatment (red lines, Figure 2). This type of sampler allows a refrigerated flask of 20 L to be filled per 24 h of the wastewater collected within a 24-h period from 8 a.m. to 8 a.m. The B7 network that drains part of the 12th and 4th districts of the city of Marseille (black framed in the RS network, Figure 2) was collected manually with a cane and bottle by the BMPM unit (Marseille Naval Fire Battalion). This district was specifically sampled because variant discrimination using multiplex RT-PCR identified an original pattern compared to the rest of the city. Both samples were transferred on ice to the NRBC laboratory (NRBC unit—nuclear, radiological, biological and chemical) of the BMPM, treated within an hour of collection for multiplex RT-PCR and then transferred directly to the University Hospital Institute Mediterranean Infection for immediate use in order to perform genome sequencing.

### 4.2. Direct Screening of SARS-CoV-2 Variants in Wastewater

Direct screening of the SARS-CoV-2 variants was performed by BMPM on 1 mL of collected wastewater previously filtered through a 5-µM Millex^®^ sterile syringe filter (SLSV025LS, Merck Millipore, MA, USA). Sample RNA was extracted using a semi-automatic eGeneup extractor and NucliSENS EasyMAG reagents (BioMérieux, Marcy l’Etoile, France). The Bio-T Kit^®^ SARS-CoV-2 UK and N501Y variants (Biosellal, BIOTK125, Dardily, France) was used for the putative detection of the Nextstrain variants 20I (V1) (UK), 20H (V2) (SA) and 20J (V3) (Br) by real-time RT-PCR (qRT-PCR) with an exogenous internal positive control (IPC). Briefly, this kit contains a ready-to-use one-step RT-PCR Master Mix allowing the detection in the same reaction well of:

− the *E gene* of Sarbecovirus, including SARS-CoV-2 with TEXAS RED labelling;− N501Y S gene mutation (shared by several variants, including the UK, SA and Br variants of SARS-CoV-2 with VIC labelling;− deletion Δ69-70 of the S gene specific to the UK variant with FAM labelling and− an exogenous internal positive control IPC RNA with Cy5 labelling. This control validates the proper conservation of nucleic acids and the absence of inhibition. An IPC control was added during the extraction of the nucleic acids.

A negative control sample (NCS) containing RNase free water replaced the sample from the extraction step. An external positive control (EPC), which was a synthetic DNA present in the kit at 10^6^ copies, replaced the sample during the RT-PCR process. This control contained specific target of SARS-CoV-2, including the N501Y mutation and SΔ69-70 deletion of the S gene. Calibration curves were also performed in parallel with the positive controls, the UK variant (IHUMI-3076), SA variant (IHUMI-3147) and Br variant (IHUMI-3191), isolated in our laboratory from nasopharyngeal swabs under previously described conditions [51] to validate the experiments.

In each well of interest, 15-µL de Master Mix and 5 µL of extracted nucleic acids (or NCS or EPC) were added. Each sample was deposited in duplicate.

The plate was sealed with an optically clear sealer; centrifuged for 20 s at 2500 rpm and the following thermal cycler parameters were used: 50 °C for 20 min, 95 °C for five minutes and 40 cycles of 10 s at 95 °C, followed by 45 s at 63 °C. The RT-PCR was carried out using the GENE-UP real-time PCR system (BioMérieux, Marcy l’Etoile, France).

For the interpretation of the control results: all RT-PCR for the NCS samples must be negative, and the RT-PCR for the EPC samples must be positive for the E gene, N501Y and SΔ69-70 deletion and positive for exogenous IPC. The Ct value obtained for the EPC samples must conform to the value indicated on the certificate of the analysis. For positive controls using known variants, the RT-PCR for the UK variant must be positive for the E gene, N501Y mutation, SΔ69-70 deletion and exogenous IPC; the RT-PCR for the SA and Br variants must be positive for the E gene, N501Y mutation and exogenous IPC.

For interpretation of the sample results: the RT-PCR for the UK variant must be positive for the E gene, N501Y mutation, SΔ69-70 deletion and exogenous IPC; the RT-PCR for the SA and Br variants must be positive for the E gene, N501Y mutation and exogenous IPC. The RT-PCR positive for the E gene only and exogenous IPC corresponded to a strain other than the UK, SA or Br variants; the RT-PCR positive for the E gene, SΔ69-70 deletion and exogenous IPC corresponded to another variant with SΔ69-70 deletion (e.g., the Denmark variant found in mink).

Using EPC control, the quantity of each variant present in the wastewater could be determined.

### 4.3. Validation of Direct Screening for SARS-CoV-2 Variants

A series of 31 SARS-CoV-2 RNAs from the isolates of known genotypes obtained in our laboratory (Appendix A) were transferred blinded to BMPM to validate the screening of the variants by the Bio-T Kit^®^ SARS-CoV-2 UK and N501Y variants (Biosellal, BIOTK125, Dardily, France).

### 4.4. Filtration and Concentration Method of Wastewater

Large particles were removed from the samples using centrifugation in six 50-mL conical centrifuge tubes at 4000× *g* for 10 min at 4 °C. Then, 250 mL of supernatant were serially filtered through a paper filter, a 5-µm polycarbonate membrane filter (TMTP04700, Merck Millipore, Burlington, MA, USA) using a Vacuum Filtration Flask, a 0.45-µM Bottle-top vacuum filtration system (514-0333P, VWR, Radnor, PA, USA) and, finally, a 0.2-µM sterile syringe filter (051733, CLEARLINE, Dutscher, Bernolsheim, France). Two 30-mL filtered samples were finally centrifuged at 100,000× *g* for two hours using a Sorvall Discovery 90SE ultra-centrifuge. Each resulting pellet was resuspended in 100 µL of PBS, and the two products were mixed to obtain a final volume of 200 µL. For evaluation of the concentration procedure, raw wastewater was directly filtered using a 5-µM Millex^®^ sterile syringe filter (SLSV025LS, Merck Millipore, Burlington, MA, USA), and 400 µL was passed to the RNA extraction.

### 4.5. RNA Extraction and RT-qPCR

The extraction of viral nucleic acids was performed using the EZ1 Virus Mini Kit (Qiagen, Hilden, Germany) following the manufacturer’s recommendations, using 200 µL of concentrated wastewater and eluted in 60 µL of elution buffer and, for the positive control for the evaluation of the concentration procedure, 400 µL of raw, 5-µM filtered wastewater and eluted in 60 µL of elution buffer. For evaluation of the SARS-CoV-2 RNA concentration, real-time PCRs were carried out specifically targeting the N-gene using the primers previously described: forward GACCCCAAAATCAGCGAAAT, reverse: TCTGGTTACTGCCAGTTGAATCTG and probe FAM: ACCCCGCATTACGTTTGGTGGACC -QSY [52]. The RT-PCR was carried out using the Superscript III Platinum One-step Quantitative RT-PCR systems with the ROX kit (Invitrogen, Waltham, MA, USA) following the manufacturer’s recommendations, with a final concentration of 400 nM of the primers and 200 nM of the probe in a final volume of 25 μL, with 2 μL of RNA. The RT-PCR programme is that described by the manufacturer. The RT-PCR were carried out using a LightCycler 480i (Roche Diagnostics, Basel, Switzerland).

### 4.6. SARS-CoV-2 Genome Sequencing

Reverse transcription was performed with 10 µL of RNA from a SARS-CoV-2-positive wastewater concentrated sample using the SuperScript VILO cDNA Synthesis kit (11754-250, Thermo Fisher, Waltham, MA, USA), according to the supplier’s protocols, using the following programme: 25 °C for 10 min, 42 °C for 120 min and 85 °C for five minutes. The cDNA of SARS-CoV-2 was then amplified 10 times by PCR with the two primer pools (ARTIC nCoV-2019 V3 Panel and 500rxn of IDT 10006788, Integrated DNA Technologies, Inc., Coralville, IA, USA) and HotStarTaq DNA Polymerase (Qiagen 203205, Hilden, Germany). The PCR reaction mixture (25 µL) per sample was prepared as follows: cDNA template (2.5 µL), primer pool (1.6-µM final concentration), dNTP (0.2-mM final concentration), 10× PCR Buffer containing 15-mM MgCl2 (10 µL for a final concentration at 1×), HotStarTaq DNA Polymerase (2.5 U per reaction) and DEPC-treated water (sufficient quantity for 25 µL). The amplification programme included an initial denaturation at 95 °C for 15 min, followed by 50 cycles, each cycle consisting of denaturation at 95 °C for 30 s, annealing at 65 °C for 30 s and extension at 72 °C for five minutes. The programme included a final extension step at 72 °C for 10 min. The PCR products were concentrated by Nucleofast 96P (Macherey Nagel ref 743100.50, Hoerdt, France) following the manufacturer’s recommendations for an elution volume of 30 µL and were separated using electrophoresis in 2% agar gel and visualised with SYBR safe (Invitrogen S3102, Waltham, MA, USA). The Monarch^®^ DNA Gel Extraction Kit (New England BioLabs, ref T1020L, Evry-Courcouronnes, France) was used for the purification of an interest band at 400 pb with an elution at 40 µL.

### 4.7. Library Preparation and Sequencing

The libraries were prepared using the Illumina COVIDSeq protocol (Illumina Inc., San Diego, CA, USA). Each previously purified PCR product was processed for tagmentation and adapter ligation using IDT for the Illumina Nextera UD Indexes Set A, B, C and D (384 indexes, 384 samples). Further enrichment and clean-up were performed as per the protocols provided by the manufacturer (Illumina Inc., San Diego, CA, USA). One COVIDSeq-positive control HT (CPC HT) and one non-template control (NTC) were added to the protocol. All libraries were pooled together. The pooled samples were quantified using Qubit 2.0 fluorometer (Invitrogen Inc., MA, USA) and fragment sizes were analysed in Agilent Fragment analyser 5200 (Agilent Inc., Santa Clara, CA, USA). The pooled libraries were further normalised to a 4-nM concentration, and 25 μL of each normalised pool containing index adapter sets were combined in a new microcentrifuge tube to a final concentration of 100 pM and 120 pM. For sequencing, the pooled libraries were denatured and neutralised with 0.2-N NaOH and 400-mM Tris-HCL (pH-8). Replicates of each of the 384 sample pools were loaded onto the S4-flow cell following the NovaSeq-XP workflow, as per the manufacturer’s instructions (Illumina Inc., San Diego, CA, USA). Dual-indexed single-end sequencing with a 36-bp read length was carried out on the NovaSeq 6000 platform.

### 4.8. SARS-CoV-2 Variant Analysis in Wastewater

For each sample, sequencing reads from both pools 1 and 2 generated by the NovaSeq device were mapped together against the SARS-CoV-2 reference genome (GenBank accession number NC_045512.2) using CLC genomics softwarev7.5 (Qiagen Digital Insights, Germany), with the default parameters. The nonsynonymous mutations present in more than 10% of the reads were taken into account. For each sample, nonsynonymous mutations were individually compared with classifying mutations that match with 30 SARS-CoV-2 variants circulating in France (Table 2). It should be noted that a given mutation could be found in different variants. In this case, all the variants harbouring this mutation were noted in the results.

### 4.9. SARS-CoV-2 Variants Repartition in Newly Infected Patients

Between 19 April and 2 May 2021—thus, three days before and ten days after the sampling—the proportion of SARS-CoV-2 variants was extracted from our laboratory anonymised database for Marseille inhabitants attending our institute to reach a SARS-CoV-2 diagnosis. All the Marseille inhabitants corresponded to what was observed in the RS wastewater network (including the B7 perimeter). In parallel, over the same period, the proportion of SARS-CoV-2 variants was extracted from our database for the inhabitants living only in the B7 perimeter. This corresponded to what was observed in the B7 network. The procedure for the routine survey using direct sequencing after RT-PCR from nasopharyngeal swabs in our institute has been previously described [51].

## Figures and Tables

**Figure 1 pathogens-10-01042-f001:**
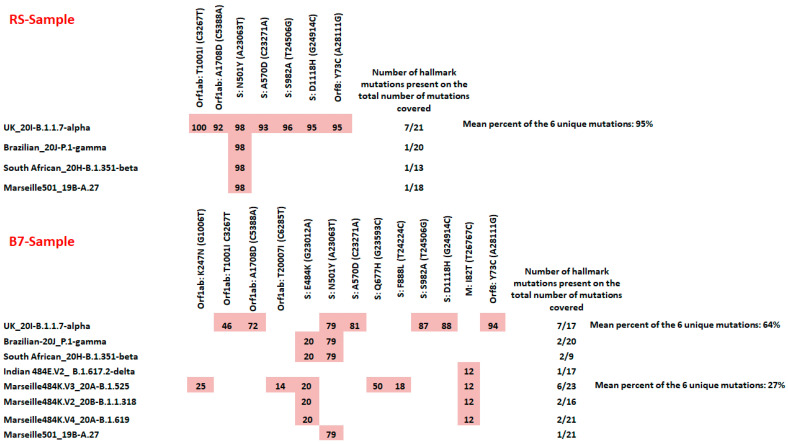
Matrix showing the pattern of mutations for the two wastewater samples. These results were provided by mapping on the SARS-CoV-2 reference genome (GenBank accession number NC_045512.2) using the CLC genomics software v7.5 (Qiagen Digital Insights, Germany), with default parameters. Only nonsynonymous mutations present in at least 10% of the reads were taken into account. Cells with a red background indicate hallmark mutations for a given SARS-CoV-2 variant. When the mutation was detected in the sewage sample, its frequency among sequencing reads is noted in each cell. Variant labels include the local nomenclature and Nextstrain, Pangolin and WHO nomenclatures.

**Figure 2 pathogens-10-01042-f002:**
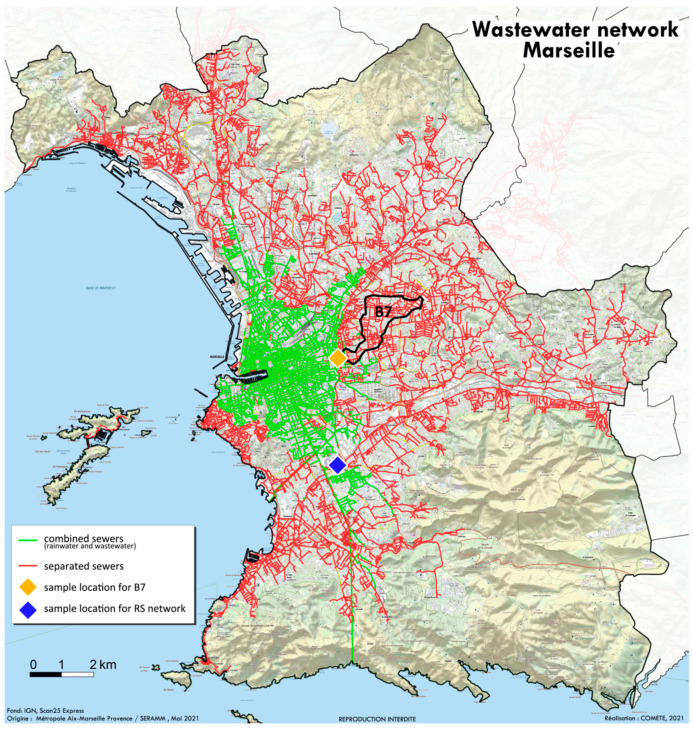
Wastewater networks in Marseille. RS network in red with its sampling point (blue square), with the B7 district identified by a black line and its sample point (yellow square).

**Table 1 pathogens-10-01042-t001:** Distribution of the SARS-CoV-2 variants in people living in Marseille (RS network) and in the B7 perimeter (B7 network) between 22 April and 2 May 2021. The results are based on typing by genome sequencing from nasopharyngeal swab samples of patients from Marseille and the B7 perimeter (1178 and 19, respectively).

SARS-CoV-2 Variant	% of Variants among People Living in Marseille (RS Network)	% of Variants among People Living in the B7 Area (B7 Network)
UK	90.5	57.9
Br	1.9	0.0
SA	1.6	0.0
B.1.525	1.7	36.9
B.1.1.138	0.4	0.0
Marseille-3	0.1	0.0
Marseille-4	0.3	0.0
20A	0.3	0.0
20B	0.2	0.0
Unclassified	3.1	5.2

**Table 2 pathogens-10-01042-t002:** Known variants of SARS-CoV-2 circulating in France.

Variant Name *	WHO Label	Classification According to Nextstrain	Pangolin Lineage
20A/15324T		20A	B.1.221
20A/20268G		20A	B.1.258
20B		20B	N.2
20D		20D	C.11
20E (EU1)		20E (EU1)	B.1.177
20F		20F	D.2
20G		20G	B.1.596
United Kingdom (UK)	Alpha	20I (V1)	B.1.1.7
Belgium		20A	B.1.214.2
Brazilian (Br)	Gamma	20J (V3)	P.1
Britain (French)		20C	B.1.616
Indian E484E.V2	Delta	21A	B.1.617.2
Indian E484Q.V1	Kappa	21A	B.1.617.1
Marseille-1		20A	B.1.416
Marseille-10		20A	-
Marseille-2		20E (EU1)	B.1.177
Marseille-3		20A	B.1
Marseille-4		20A.EU2	B.1.160
Marseille-452R-19B		19B	A.21
Marseille-484K.V1		20B	R.1
Marseille-484K.V2		20B	B.1.1.318
Marseille-484K.V3	Eta	21D	B.1.525
Marseille-484K.V4		20A	20A
Marseille-5		20C	B.1.367
Marseille-501		19B	A.27
Marseille-6		20A	B.1
Marseille-7		20A	-
Marseille-8		20B	B.1.1.269
Marseille-9		20B	B.1.1.241
South African (SA)	Beta	20H (V2)	B.1.351

* Marseille variants are local variants with discrete sequence modifications.

## Data Availability

Data is contained within the article and Appendix A.

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
