# Peer review of "Monitoring the Circulation of SARS-CoV-2 Variants by Genomic Analysis of Wastewater in Marseille, South-East France"

_pathogens, 2021, doi:10.3390/pathogens10081042_

Round 1

Reviewer 1 Report

SARS-CoV-2 as well as its variants have spread all over the world and caused enormous problems. As SARS-CoV-2 is likely to keep mutating and spreading, it is critical to establish a surveillance system to keep track of not only the existence but also the variant types. Current surveillance system is largely based on symptomatic (or even hospitalized) patients, and thus may be biased. In this study, the authors developed methods to detect the variant types of viruses using wastewater samples. They compared the two detection methods: viral genome sequencing, and a commercial kit based on RT-PCR. This study is important to the field. However, this manuscript is very hard to follow due to the ambiguous figure legends and numbers in the context – different methods were applied on various samples, which need to be specified to clarify. For example:

  1. It is not shown in the figure legend of Figure 1, that which method was used to generate these results.

  1. Amino acid numbers are used throughout the manuscript, but nucleotide numbers are used in Figure 1. Can the corresponding amino acid numbers be shown, too?

  1. Section 2.3, “The proportion of UK variants observed in the population (95%) …”. Does that refer to genome sequencing results for patient samples instead of waste water samples?

  1. Section 2.3, “… was closer to that of sewage genome sequencing (95%)…”. Is this number corresponding to the average value of the RS-B.1.1.7 row? If yes, does it mean that the patient genome sequencing result was compared to the RS wastewater sample but not the B7 sample? But the later part of this sentence “than that obtained using the Bio-T Kit125® screening test (70%)” refers only to the RS sample according to Table 1. This is a major result of this study but very hard to follow.

  1. Section 2.3, “For the RS network, the proportion of UK variants observed in B7 (58%) was more in line with that of sewage genome sequencing (64%) than that obtained using the Bio-T Kit125® screening test (8%).” Does it refer to the RS sample or the B7 sample?

  1. Table 2. Is this table presenting patient genome sequencing results?

Additional comments:

  1. Country/city names were used to represent variants throughout the manuscript. But it would be good to use their scientific names.

  1. Table 3. “Indian E484K.V2” was used to refer to the Delta version, which does not contain the E484K mutation.

  1. It is surprising that there is a big difference in terms of variant types between the RS and B7 samples, although these samples were extracted from a same city. Can the authors discuss about this?

  1. Can the author discuss about possible reason that caused the difference of the results of the wastewater samples between genome sequencing and the RT-PCR kit?

  1. Would variant identification results of the wastewater samples largely depending on the stool waste? If yes, it might be useful to discuss the possibility that the viral amount of different variants may not be equally present in stool waste, and may need to be normalized to be better represent the percentage of variants in population.

  1. Page 4, line 1: please change “B.1.617.V2” to “B.1.617.2”.

Author Response

Reviewer :It is not shown in the figure legend of Figure 1, that which method was used to generate these results.

Authors : To generate these results, we used a mapping on the SARS-CoV2 reference genome provided by the CLC genomics software. This tool provides all the positions with mutations by comparing with the reference genome. We added the method used in the figure legend of the figure 1, as required:

“These results were provided by mapping on the SARS-CoV-2 reference genome (Genbank accession number NC_045512.2) using the CLC genomics software (https://digitalinsights.qiagen.com/), with default parameters. Only non synonymous mutations present in at least 10% of the reads were taken into account.”

Reviewer :Amino acid numbers are used throughout the manuscript, but nucleotide numbers are used in Figure 1. Can the corresponding amino acid numbers be shown, too?

Authors : The figure 1 has been corrected as required by adding the corresponding amino acids numbers for each position indicated in the first raw.

Reviewer :Section 2.3, “The proportion of UK variants observed in the population (95%) …”. Does that refer to genome sequencing results for patient samples instead of waste water samples?

 Authors :This was a mistake, UK in patients is 90% (not 95%) as presented in table 1. The legend of table 1 was tentatively improved

Reviewer :Section 2.3, “… was closer to that of sewage genome sequencing (95%)…”. Is this number corresponding to the average value of the RS-B.1.1.7 row? If yes, does it mean that the patient genome sequencing result was compared to the RS wastewater sample but not the B7 sample? But the later part of this sentence “than that obtained using the Bio-T Kit125® screening test (70%)” refers only to the RS sample according to Table 1. This is a major result of this study but very hard to follow.

 Authors : yes, 95% the average value based on the 6 unique mutations based on unique mutations. I agree that is was a little bit confusing. We add a last column were mean percentage of unique mutation is given for Rs and for B7 in figure 1.

Reviewer :Section 2.3, “For the RS network, the proportion of UK variants observed in B7 (58%) was more in line with that of sewage genome sequencing (64%) than that obtained using the Bio-T Kit125® screening test (8%).” Does it refer to the RS sample or the B7 sample?

 Authors : I agree that this sentence was not clear! It was modified.

Reviewer :Table 2. Is this table presenting patient genome sequencing results?

 Authors :Yes (now table 1 as reviewer 2 asked for removing table 1). We have modified table legend for clarification

Reviewer :Country/city names were used to represent variants throughout the manuscript. But it would be good to use their scientific names.

 Authors : I agree that the classification is increasingly complex. This is the reason we presented variants in table 3 in order that each reader can do conversion.

Reviewer :Table 3. “Indian E484K.V2” was used to refer to the Delta version, which does not contain the E484K mutation.

 Authors : We thank the reviewer to highlight this mistake; we corrected it. The Indian variant we wanted to refer was the Indian E484E.V2 variant.

Reviewer :It is surprising that there is a big difference in terms of variant types between the RS and B7 samples, although these samples were extracted from a same city. Can the authors discuss about this?

 Authors : we have only hypotheses about this that were presented in discussion

Reviewer :Can the author discuss about possible reason that caused the difference of the results of the wastewater samples between genome sequencing and the RT-PCR kit?

 Authors : We believe that some additional mutations around the del69-70 region for some of these strains may prevent recognition of the probe. The targeted region is known (del69-70), but the primers and the probes of the kit being kept secret by the manufacturer. This prevents us from verifying this hypothesis.

Reviewer :Would variant identification results of the wastewater samples largely depending on the stool waste? If yes, it might be useful to discuss the possibility that the viral amount of different variants may not be equally present in stool waste, and may need to be normalized to be better represent the percentage of variants in population.

Authors : We totally agree with that and tried to discuss that at the end of the discussion, in spite that we believe it could be very complex to explore.

Reviewer :Page 4, line 1: please change “B.1.617.V2” to “B.1.617.2”.

Authors :

This is corrected

Reviewer 2 Report

The manuscript presents the results of a study on the monitorization of SARS-CoV-2 variants by analyzing the wastewater in Marseille, France.  This method proved to be a simple and unbiased way to evaluate the evolution of the epidemics in a limited region/city.

The paper is well written. The Abstract presents the most important aspects of the study. In the Introduction, the authors present in a clear way the field of the SARS-CoV-2 pandemic and the use of detection of the virus in wastewater in order to evaluate the epidemiologic evolution in an area.

Regarding the presentation of the results, the authors present some results in both ways, in text, and in tables. It would be better to choose one way, for example, to present the data from Table 1 in the text and to use the tables (Table 2 and Figure 1), and to reduce the corresponding part of the text.

The Discussion presents an analysis of the results in correlation with data from other papers. I consider that the authors should include one last paragraph as a Conclusion. 

The Methods are presented very clear, including all the steps of the study. 

The references included, as it is supposed to be, most of the titles from the last 2 years.  

Some minor editing checks should be done (for example, World with a capital initial letter - in introduction; the UK as abbreviation first time should be presented in full, RT and B7 are explained in the methods, but after their first use in the text).

Author Response

Reviewer : The manuscript presents the results of a study on the monitorization of SARS-CoV-2 variants by analyzing the wastewater in Marseille, France.  This method proved to be a simple and unbiased way to evaluate the evolution of the epidemics in a limited region/city.

Reviewer :The paper is well written. The Abstract presents the most important aspects of the study. In the Introduction, the authors present in a clear way the field of the SARS-CoV-2 pandemic and the use of detection of the virus in wastewater in order to evaluate the epidemiologic evolution in an area.

Authors : thank you

Reviewer :Regarding the presentation of the results, the authors present some results in both ways, in text, and in tables. It would be better to choose one way, for example, to present the data from Table 1 in the text and to use the tables (Table 2 and Figure 1), and to reduce the corresponding part of the text.

Authors : we removed table 1 as suggested by reviewer and presented the associated data in the text only. We preferred save both text and figure 1 and table 1 (former table 2) as the data presented are complex to present using only figure or only text.

Reviewer :The Discussion presents an analysis of the results in correlation with data from other papers. I consider that the authors should include one last paragraph as a Conclusion. 

Authors : We add a conclusion paragraph as suggested by adding a summary sentence et replacing the last part of the text of the discussion which was a kind of conclusion.

Reviewer :The Methods are presented very clear, including all the steps of the study. 

Authors : Thank you

Reviewer :The references included, as it is supposed to be, most of the titles from the last 2 years.  

Authors : OK

Reviewer :Some minor editing checks should be done (for example, World with a capital initial letter - in introduction; the UK as abbreviation first time should be presented in full, RT and B7 are explained in the methods, but after their first use in the text).

 Authors : We have tried to make these corrections. For RS and B7 this is the disadvantage of newspapers which ask to present the materials and methods at the end of the article. We hope the changes we made are appropriate.

Round 2

Reviewer 1 Report

I do not have additional comments.